# Pure non-local machine-learned density functional theory for electron correlation

Johannes T. Margraf [1]✉ & Karsten Reuter[1,2]

Density-functional theory (DFT) is a rigorous and (in principle) exact framework for the description of the ground state properties of atoms, molecules and solids based on their electron density. While computationally efficient density-functional approximations (DFAs) have become essential tools in computational chemistry, their (semi-)local treatment of electron correlation has a number of well-known pathologies, e.g. related to electron self-interaction. Here, we present a type of machine-learning (ML) based DFA (termed Kernel Density Functional Approximation, KDFA) that is pure, non-local and transferable, and can be efficiently trained with fully quantitative reference methods. The functionals retain the mean-field computational cost of common DFAs and are shown to be applicable to non-covalent, ionic and covalent interactions, as well as across different system sizes. We demonstrate their remarkable possibilities by computing the free energy surface for the protonated water dimer at hitherto unfeasible gold-standard coupled cluster quality on a single commodity workstation.

---

[1] Chair for Theoretical Chemistry and Catalysis Research Center, Technische Universität München, Lichtenbergstraße 4, D-85747 Garching, Germany. [2] Fritz-Haber-Institut der Max-Planck-Gesellschaft, Faradayweg 4-6, D-14195 Berlin, Germany. ✉email: johannes.margraf@ch.tum.de

In their seminal 1964 paper, Hohenberg and Kohn proved that there exists a universal functional $F[\rho]$ of the electron density $\rho$, that captures all electronic contributions to the total energy of a system of interacting electrons[1]. This universal functional has since become something of a holy grail for chemistry, physics, and materials science, as its knowledge would allow determining the exact ground-state energy and electron density for any molecule or solid[2]. Unfortunately, the concrete form of $F[\rho]$ itself has remained elusive. Indeed, it has been shown that the universal functional likely has the same prohibitive computational complexity as solving the Schrödinger equation directly[3].

Nevertheless, Hohenberg and Kohn's density functional theory (DFT) has become an essential method in the toolkits of computational chemistry, condensed matter physics, or surface science. This is mostly owing to the formulation of Kohn and Sham (KS), which reduces the problem to finding a density functional $E_{xc}[\rho]$ for electronic exchange and correlation[4]. Again, the exact form of $E_{xc}[\rho]$ is unknown, but many useful density functional approximations (DFAs) exist, which are generally considered to offer a good trade-off between computational complexity and accuracy.

The large zoo of DFAs that has been developed over the years is often organized in the hierarchy of Jacob's ladder, where approximations are grouped according to the ingredients that are used in their functional form[5]. At the lower rungs of this ladder, (semi-)local DFAs are found, which only require information about the local density and its derivatives. Such functionals are sometimes called pure, because they can be computed from the electron density alone. At higher-rungs, information about the occupied and/or unoccupied KS orbitals is also included[6–8]. These so-called (double) hybrid DFAs are therefore no longer pure in the sense described above. This increases their computational complexity, but also makes them more accurate, because they incorporate non-local information.

In spite of their known limitations (e.g., regarding electron self-interaction)[9], the pure functionals at the bottom of Jacob's latter are a widely used state-of-the-art, e.g., in practical calculations of large systems or with extensive sampling. This reveals a crucial dilemma of Jacob's ladder, namely that very often it is not possible to use a higher-rung functional due to computational constraints, even if the nature of the system of interest would in principle require a more accurate description.

Recently, machine-learning (ML)-based DFAs have been shown to break the constraints of Jacob's ladder, offering highly accurate, pure, and non-local density functionals for different one-dimensional model systems[10–13]. Although these approaches show that it is possible to learn the highly non-linear mapping from the electron density to the energy from data, they cannot be directly transferred to real systems. A straightforward real-space representation of the electron density (e.g., on a grid) is extremely inefficient in three dimensions. Even for a small molecule, a single reference energy value would have to be fitted as a function of on the order of one hundred thousand input values. Furthermore, grid-based models are not in general size-extensive, particularly if they require the grid to be of constant size. Very recently, Burke and co-workers developed ML-based DFT models, which represent the density in a plane-wave basis[14,15]. These models have successfully been applied to real molecules, though they are still not size-extensive.

These issues can be circumvented if the reference energy is projected onto the grid in the form of an energy density[16–18]. This brings the number of target values to the same order of magnitude as the number of input values, leading to a much better defined fitting problem. On the other hand, this also makes the resulting ML-DFAs more like the traditional functionals in Jacob's ladder. For instance, if a semilocal Ansatz is chosen to fit a reference energy density, the resulting functional will display

electron self-interaction, even if the reference method does not[17]. Developing correlation functionals in this manner is particularly challenging. Correlation energy densities based on high-level wavefunction methods can, e.g., display significant positive values at the centers of stretched bonds, where the electron density is vanishingly small[19].

In this paper, we therefore propose a new route to pure ML-based DFAs, using a density-fitting representation of the electron density. This representation is much more compact than a real-space grid, and allows decomposing the density into atomic contributions. As a consequence, the presented ML-DFAs can readily be applied to real molecules and are by construction size-extensive. At the same time, any reference method can be used for training without the need for real-space projection or energy decomposition.

## Results

**Density representation**. In KS-DFT, the electron density is computed from the occupied KS orbitals $\psi_i(r)$. These are in turn expanded as a linear combination of basis functions $\chi_\mu(r)$:

$$\rho(r) = \sum_i |\psi_i(r)|^2 = \sum_{\nu\mu} D_{\mu\nu}\chi_\mu(r)\chi_\nu(r) \qquad (1)$$

with the density matrix elements $D_{\mu\nu}$. A noteworthy aspect of eq. (1) is that the density is expanded in terms of products of basis functions, which are not in general atom-centered, even if the basis functions themselves are (see Fig. 1, left). This leads to the appearance of memory intensive four-index integrals in the computation of Coulomb and exchange contributions to the KS matrix.

In many electronic structure codes it is therefore common practice to use additional density-fitting (DF) basis functions $\phi_Q(r - r_A)$, which allow an atom-centered expansion of the density (see Supplementary Note 1):

$$\rho(r) = \sum_A \sum_Q C_Q^A \phi_Q(r - r_A) = \sum_A \rho_A(r). \qquad (2)$$

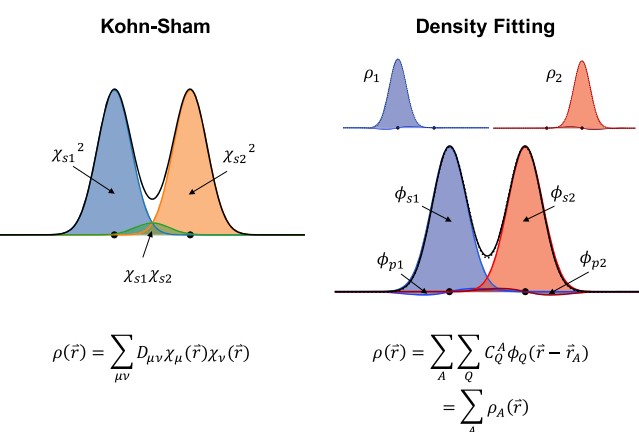

**Fig. 1 Illustration of conventional and density-fitting based basis expansions of the electron density.** Left: In a conventional Kohn–Sham DFT calculation, the electron density (solid black line) is expanded in terms of density matrix elements $D_{\mu\nu}$ and products of basis functions $\chi_\mu\chi_\nu$. Right: Density-fitting (DF) allows expanding the density in terms of fitting coefficients $C_Q^A$ and atom-centered basis functions $\phi_Q$ (dotted black line). The DF expansion can unambiguously be decomposed into atomic contributions. Note that higher angular momentum functions are needed in the DF basis to correctly describe the overlap region between the atoms. This is illustrated in the schematic figure by the use of only s-type basis functions for the Kohn–Sham expansion and s- and p-type basis functions for the DF expansion.

Here, the first sum is over all atoms $A$, and $C_Q^A$ is the expansion coefficient of basis function $Q$ on atom $A$. Unlike eq. (1), the DF expansion can readily (and unambiguously) be decomposed into atomic densities $\rho_A$ (see Fig. 1, right)[20,21].

It should be noted that the accurate DF representation of $\rho(r)$ requires significantly larger basis sets than those used for the expansion of the KS orbitals. Nonetheless, the density can typically be precisely represented with the knowledge of ca. 100 coefficients per atom and a set of DF basis functions that only depend on the element of the atom. This is in contrast to eq. (1), where the product basis is dependent on the geometry (i.e. on the relative position of pairs of atoms) of the molecule in question.

The DF basis thus offers a highly compressed and transferable representation of $\rho$, both of which are desirable properties for ML. However, there is also a significant downside to this representation, namely that the coefficients $C_Q^A$ are not in general rotationally invariant. In other words, rotating a molecule and its density will lead to a different set of coefficients $C_Q^A$, even though the target energy remains unchanged. Although this invariance could in principle be learned from data, this would require significantly more training data and only lead to an approximately invariant model.

This issue can be circumvented by borrowing a trick from the Smooth Overlap of Atomic Positions (SOAP) representation of atomic environments[22]: Instead of using the coefficients directly, we use their rotationally invariant power spectrum $\mathbf{p_A}$ (see Supplementary Note 2), which has the added benefit of further compressing the representation. In the following, we will refer to this as the rotationally invariant density representation (RIDR).

**Correlation functionals from ML.** We can now proceed to construct a ML-DFA based on the RIDR. Herein, we will focus on learning correlation energy functionals from calculated Hartree–Fock (HF) densities. Although exchange is treated exactly within HF, the combination of full HF exchange with conventional pure DFT correlation functionals leads to poor results[23]. This is because the non-local exchange in HF is incompatible with (semi-)local correlation[5]. We therefore train our model on non-local correlation energies obtained from many-body wavefunction methods. Specifically, we will consider second-order Møller–Plesset theory (MP2) and gold-standard Coupled Cluster theory with single, double, and perturbative triple excitations (CCSD(T)), to illustrate the approach for both an efficient and a fully quantitative treatment of electron correlation, respectively[24–26]. In addition to being non-local, both these methods also provide much improved fractional charge behavior, overcoming one of the main pathologies of conventional DFAs[9,27–29].

Because wavefunction-based reference calculations are computationally expensive, it is critical to choose an ML approach that is as data-efficient as possible. In chemistry applications, Kernel Ridge Regression (KRR) has been found to be a good choice in this respect[30–33]. We can write a generic KRR correlation functional as:

$$E_c[\rho] = \sum_i^N \alpha_i K(\rho, \rho_i),  \qquad (3)$$

where the sum runs over $N$ training densities, $\alpha_i$ are regression coefficients and $K(\rho, \rho_i)$ is a kernel function that measures the similarity between the target and training densities. To ensure size-extensivity of the functional, we construct $K(\rho, \rho_i)$ from the atomic densities provided by the DF representation (see eq. (2)):

$$K(\rho_i, \rho_j) = \sum_{A \in i, B \in j} k(\rho_A, \rho_B). \qquad (4)$$

Now a kernel $k(\rho_A, \rho_B)$ that measures the similarity of atomic densities can be defined using the RIDR. A wide range of kernel functions are possible, but we found that a simple polynomial Ansatz (as used for the SOAP kernel) already displays very promising performance, as discussed below:

$$k(\rho_A, \rho_B) = \left( \frac{\mathbf{p_A}^\top \mathbf{p_B}}{\sqrt{(\mathbf{p_A}^\top \mathbf{p_A})(\mathbf{p_B}^\top \mathbf{p_B})}} \right)^2. \qquad (5)$$

Having defined the kernel function, the regression coefficients $\alpha_i$ (in eq. (3)) can be determined in closed form, for a given training set (see Supplementary Note 3). In the following these functionals are referred to as kernel density functional approximations (KDFA).

**Performance and applications.** KDFAs need to be trained on data, and are consequently always to some extent system specific. Nonetheless, a useful ML-DFA methodology should be general in the sense that it can easily be trained for different chemistries and system sizes[34]. Herein, three types of systems are considered, namely water clusters, microsolvated protons and linear alkanes of different sizes. In all cases, training and test structures are randomly drawn from molecular dynamics (MD) simulations at 350 K (see Methods section for details). For this part, we will concentrate on the results obtained with MP2 reference data. As also further commented below, CCSD(T) energies can be learned with the same accuracy and a full account of these results is provided in the SI.

In Fig. 2, learning curves of ML correlation functionals for the three types of systems are shown. In all cases, mean absolute errors (MAEs) below 25 meV (roughly corresponding to $k_B T$ at 300 K) are reached with 100 training examples. Indeed, in many cases so-called chemical accuracy (1 kcal mol$^{-1}$ ≈ 43 meV) is already achieved with only 10 training structures. Within each class, the MAEs generally increase for larger systems. However, the errors for pure and protonated water clusters with three and four molecules are nearly identical. This is readily understood, as the intermolecular interactions are approximately additive in these systems. In contrast, the interactions in longer alkane chains are harder to learn, e.g., owing to more pronounced interactions between the molecular termini in butane ($C_4H_{10}$) than in propane ($C_3H_8$).

Overall, these results show that highly accurate, non-local correlation functionals can be learned from easily tractable reference data sets. In fact, the similarity of the learning curves obtained for the three systems suggests that high-level energies for a large ensemble of configurations can already be obtained with only 10–100 reference calculations. In many applications, even such a limited number of high-level reference calculations may not be feasible, however, owing to the prohibitive scaling of wavefunction methods with system size[35,36].

Here, the size-extensivity of our approach becomes important, as it enables us to train on small systems and predict the energies of larger ones[34,37]. In Fig. 3, this is highlighted for the water octamer and octane, with the corresponding KDFAs trained exclusively on the systems with up to four monomers (i.e., at most half the size of the predicted systems). Clearly, this is a challenging test for the transferability of the functionals. The observed MAEs are indeed somewhat larger than for models validated on configurations of the same size as the training set (74 and 48 meV for $(H_2O)_8$ and octane, respectively). A larger error should be expected for larger systems, however, precisely owing to the extensive nature of the correlation energy[38,39]. Furthermore, the prediction errors are somewhat systematic, in particular for $(H_2O)_8$. Consequently, the MAE for relative energies of water

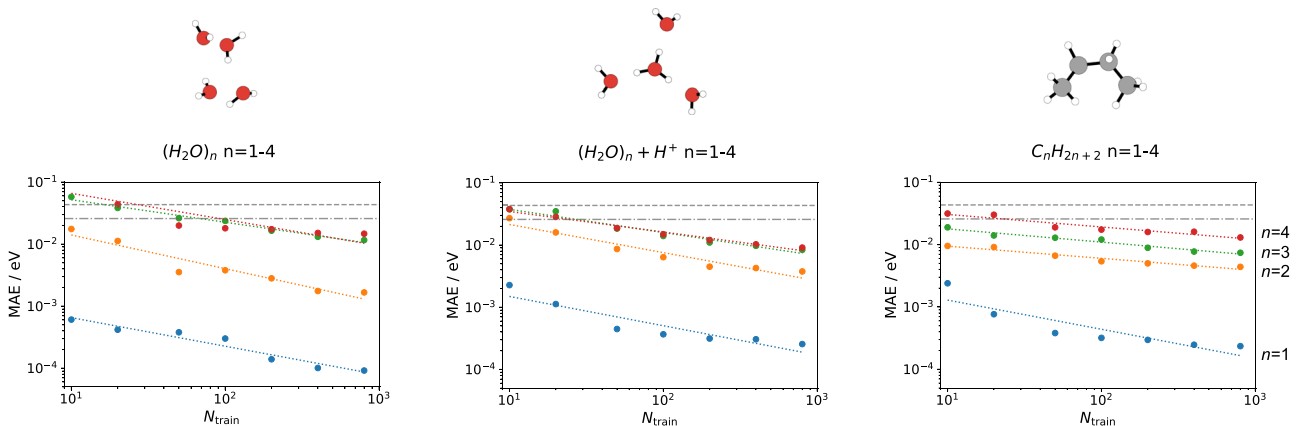

**Fig. 2 Learning curves.** Shown is the mean absolute error (MAE) vs number of training densities ($N_{train}$) for the correlation energy in water clusters (left), protonated water clusters (center) and alkane configurations (right). For reference, the dashed line denotes an error of $1\,kcal\,mol^{-1} \approx 43\,meV$ and the dash-dotted one an error of $k_B T \approx 25\,meV$ (at 300 K).

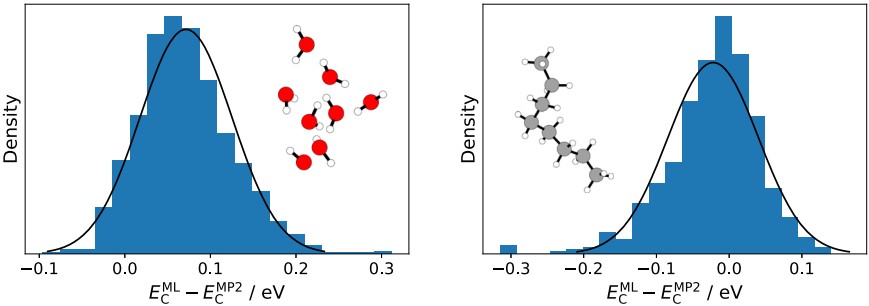

**Fig. 3 Model transferability.** Shown are error histograms for predicted correlation energies $E_C^{ML}$ of water octamer (left) and octane configurations (right) relative to MP2 reference energies $E_C^{MP2}$. The corresponding models were exclusively trained on systems with up to four monomers, exploiting the size-extensivity of the ML correlation functional.

octamer configurations is further reduced to 42 meV (46 meV for octane).

Of comparable computational cost as conventional low-rung functionals, the transferable KDFA approach provides access to quantitative CCSD(T) simulations for systems that before were on the brink of largest-scale supercomputing studies or simply not tractable at all. We demonstrate this for the shared proton in a protonated water dimer, as an intensely studied testbed for the role of electron correlation and nuclear quantum effects in hydrated excess protons[40–49]. An ensemble of 100,000 uncorrelated configurations of the protonated water dimer was drawn from a 5 ns MD trajectory generated at the semiempirical GFN2-XTB level of theory[50]. From this ensemble, 100 structures were randomly sampled and their energies computed at the CCSD(T)/def2-TZVP level. These configurations were used to train a KDFA, which was used to predict the energies of all 100,000 configurations. These KDFA energies were then used to generate a CCSD(T)-level ensemble, via the Monte Carlo resampling (RSM) approach of Essex and co-workers (see Supplementary Note 4)[51].

The two dominant reaction coordinates for this system are the oxygen–oxygen distance ($r_{OO}$) and the proton transfer coordinate $v$[40]. The latter is defined as the difference between the O-H distances of the shared proton and each oxygen atom, with a value of $v = 0$ meaning that the proton is equidistant to both oxygens, whereas positive or negative values indicate that the proton is closer to one of the oxygen atoms. In Fig. 4 (left) the free-energy surface derived from the semiempirical MD trajectory is shown with respect to these coordinates. This reveals a fairly broad single-well potential with a minimum around $r_{OO} \approx 2.45$ Å

and $v = 0$. The location of the minimum and the overall shape of the well are in good agreement with previously reported probability distributions obtained from MD simulations with dispersion-corrected (semilocal) DFT[52].

In contrast, the free-energy surface for the CCSD(T)-quality KDFA displays some strikingly different features. Here, the minimum is more narrow and at shorter $r_{OO}$. Furthermore, the potential well has a distinct heart-shape, meaning that at larger values of $r_{OO}$, the proton is no longer equidistant to both oxygen atoms but preferentially located closer to one or the other. Finally, the energy range of both free-energy plots differs by over 100 meV, indicating that the minimum is much deeper and narrower at the CCSD(T) level. The comparison with the semiempirical surface (and analogous semilocal DFT surfaces)[40,52] shows that the electron delocalization errors of these methods also leads to proton over-delocalization. Interestingly, the inclusion of nuclear quantum effects actually does delocalize the proton more strongly[40,41]. However, to obtain good agreement with the experimental properties of water both an explicit treatment of nuclear quantum effects (i.e., via path-integral MD) and a quantitatively correct classical potential energy surface (as provided by our CCSD(T)-quality KDFA) are required[53,54].

Overall, the features of the presented CCSD(T) surface are in good agreement with the one recently reported by Kühne and co-workers, which was based on tour-de-force MD simulations requiring millions of CCSD/cc-pVDZ calculations[41], i.e., in comparison with our work without perturbative triples in the coupled cluster ansatz and using an inferior basis set. In contrast, our combined KDFA/RSM approach only required 100 CCSD(T) reference calculations and was quickly performed in a matter of

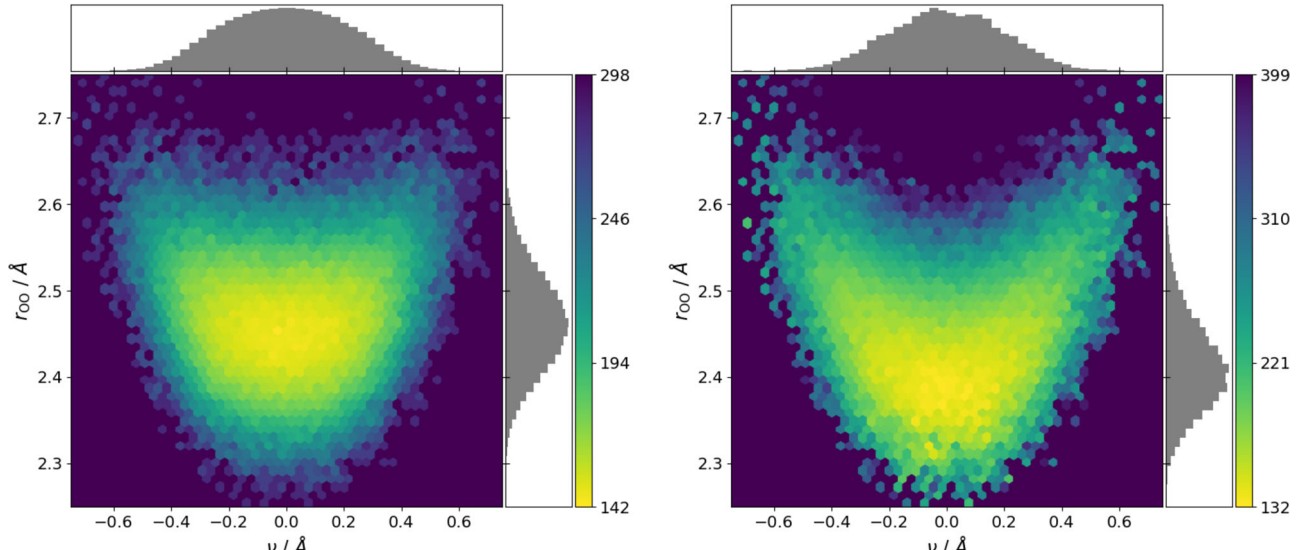

**Fig. 4 Free-energy surfaces for the shared proton in a protonated water dimer.** Left: original surface from the semiempirical molecular dynamics simulation. Right: resampled free-energy surface from the machine-learned coupled cluster functional. Histograms of the underlying distributions along both coordinates are also shown. Free energies are given in meV as a function of the proton reaction coordinate ($\nu$) and the O–O distance ($r_{OO}$). Note the different topology of the two surfaces discussed in the text, but also the largely different energy range obtained at the two levels of theory.

hours on a single workstation, even though we chose to employ a much larger ensemble to arrive at a better free-energy sampling.

## Discussion

Given the large variety of ML approaches reported in recent years, it is important to discuss the proposed KDFA method in a broader context. Most prominently, kernel and neural network regression has been very successfully used to directly learn the relationship between atomic coordinates and ground-state energies, i.e., the potential energy surface (PES)[22,30,31,33,48,55–66]. The main advantage of this direct approach is that electronic structure calculations are only required for training, whereas predictions can be performed using just the atomic coordinates as input. On the other hand, this also means that the complex physics underlying the PES has to be learned completely from data, which often translates to very large training sets from tens of thousands to millions of configurations, depending on the system (see Supplementary Note 5)[67,68]. Another downside is that most such ML forcefields are by construction short-ranged, meaning that long-range Coulomb and dispersion interactions are not included.

Von Lilienfeld and co-workers proposed the so-called Δ-ML approach to mitigate these downsides[69]. Here, the idea is to use an inexpensive semiempirical method as a baseline and learn the energy difference to a higher-level method like DFT or CC. They showed that this requires much less training data to reach a given accuracy than for a pure ML model. Broadly speaking, the KDFA approach proposed here also falls into this category. However, rather than learning the total energy differences between different approximations wholesale, we exclusively focus on the correlation energy, which is generally much smoother (and therefore easier to learn) than the total PES[32,70]. Furthermore, unlike the original Δ-ML approach, we use the electron density as input, rather than geometric information. Our ML models are thus genuine density functionals.

The presented approach is closest in spirit to the work of Miller and co-workers, where pair-correlation energies are learned from molecular orbital based descriptors (MOB-ML)[34,37]. As in their work, our models are based on a rigorous physical theory of electron correlation and are by construction size-extensive. The main difference is that in our case no previous energy decomposition or orbital localization is required. This enables us to choose arbitrary reference methods, for which pair-correlation energies may not be available or implemented (i.e., the random phase approximation, RPA, or quantum Monte Carlo methods). Furthermore, the requirement of orbital localization makes the application of MOB-ML to metallic or small band-gap systems fundamentally difficult.

Of course, the work of Burke, Müller, Tuckermann, and co-workers on ML-based DFT is also highly pertinent[10,11,13–15]. Their most recent method focuses on the integrated prediction of both the valence electron density and the exchange-correlation energy[14]. This work was originally aimed at accelerating the computation of established DFAs, but has very recently been extended to also learn corrections to higher-level methods like CCSD(T)[15]. The main difference to our work is their choice of density representation. Specifically, they work in a plane-wave basis instead of the atom-centered Gaussians used herein. This makes learning the density much easier, since all basis functions are orthogonal. As a consequence of this choice, their models are not size-extensive, however. Furthermore, molecular symmetry can only be included in an approximate form, i.e., through data-augmentation. Consequently, these models can only be applied if at least some number of high-level CC calculations are affordable for the complete system, and if the predictions are done on fairly similar configurations (i.e., within an MD trajectory). It should also be noted that Ceriotti, Corminbeauf, and co-workers have recently developed a method for learning electron densities in atom-centered basis sets[20,21,71]. In future work, integrated, size-extensive KDFAs could thus be developed that can predict both densities and energies.

Finally, a note on self-consistency is in order. KS-DFT calculations are commonly carried out self-consistently. One could therefore argue that the presented approach is more closely related to post-HF methods like Coupled Cluster theory. Nonetheless, we feel that our models are more accurately described as DFAs, as they exclusively depend on the electron density. In particular, no information related to virtual orbitals is required. The models therefore share the favorable computational scaling of pure DFAs. We also note that higher-rung DFAs (i.e., double

hybrids or RPA-based functionals) are still commonly used non-self-consistently[6,8]. Indeed, even classic semilocal functionals like Becke's B88 were developed in a non-self-consistent framework[72]. In the same vein, a self-consistent extension of the proposed method is clearly possible and will be the subject of future work. This will, however, likely require significantly more training data.

To conclude, we have presented a novel method to learn pure, non-local and transferable density functionals for the correlation energy from data. The high accuracy and data-efficiency of the method was demonstrated for a range of non-covalent, ionic and covalent systems of different sizes. In all cases, MAEs below 25 meV could be achieved with <100 training structures. We also demonstrated the transferability and size-extensivity of the approach by training and predicting on differently sized systems. As an exemplary application of the new method, a highly accurate free-energy surface for the shared proton in a protonated water dimer was obtained by resampling a semiempirical MD trajectrory with a CCSD(T)-based KDFA at mean-field computational cost.

## Methods

**Computational details**. Most electronic structure calculations were performed with the `Psi4` package[73]. The KDFA method was implemented as an external plugin to `Psi4` via the `Psi4numpy` interface[74]. CCSD(T) and MP2 calculations were performed with `ORCA`[75]. The Karlsruhe def2-TZVP basis set and corresponding DF basis were used throughout[76,77].

**Molecular dynamics**. NVT MD simulations were performed with the semi-empirical GFN2-XTB method and the atomic simulation environment[50,78]. All simulations were run with a timestep of 0.5 fs and a Langevin thermostat with a coupling constant of 0.1 atomic units. For the learning curves in Fig. 2, decorrelated snapshots from 50 ps trajectories at 350 K were used. For the free-energy surfaces in Fig. 4, a trajectory of 5 ns at 300 K was calculated.

## Data availability

All data sets used in this paper are available in the supplementary information (Supplementary Data 1).

## Code availability

The code used to fit the ML models is available at https://gitlab.com/jmargraf/kdf.

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

## Acknowledgements

We acknowledge funding from the German Research Foundation (DFG) through its Cluster of Excellence *e*-conversion EXC 2089/1.

## Author contributions

J.T.M. and K.R. devised the project. J.T.M. wrote the KDFT code and performed all calculations. All authors contributed to writing the manuscript.

## Funding

## Competing interests

The authors declare no competing interests.
