## [Peer Review File · Nature Communications]

REVIEWER COMMENTS

Reviewer #1 (Remarks to the Author):

The method describes a solid piece of work with promising results. I am supportive of its publication. However, I do have concerns about whether the method is not a sufficient improvement upon existing methods and whether the numerical examples are sufficiently compelling to merit publication in Nat Commun.

Reviewer #2 (Remarks to the Author):

In the manuscript the authors discuss a machine learning method to accurately calculate the correlation energy from Hartree-Fock densities. The methodology is new, innovative, and accurate. This work is certainly worthy of publication; however, additional work may be required to merit publication in Nature Communications.

Overall concerns:

1. The authors are not performing DFT calculations – their methodology is significantly more similar to post-HF methodologies. Particularly since they use the HF density. This methodology offers a way to get a correlation energy, like MP2 or CCSD(T). I think if the authors are going to motivate this work as a route to DFA, they need to show a methodology that would be self-consistent with the electron density. Right now, the methodology is a post-HF method, and it should be represented as such early in the paper. I understand that density (or the density fitted basis) is the connection to the correlation energy. But right now, the difference between what is said in the abstract and what is actually done in the paper is large in my opinion.
2. The first results they show are on water clusters and alkane chains. While they show “transferability” for correlation energy learned on propane and applied on butane and suggest that this makes it applicable to large systems they don’t show any larger systems. Since they are comparing to MP2, getting reference data for larger chain (even if they are not training on it) should be very feasible. I think a much stronger show of transferability would be to show significant longer alkane chains, octane for instance. Right now, I don’t believe the authors should claim they have a transferable density functional if they can only show it can transfer from 3 water molecules to 4 and propane to butane. Even better show of transferability would be between the same atom in different environments (for instance ethane and ethene)
3. They only show results for 3 systems, water clusters, alkane chains, and a proton transfer. And they only truly have reference data for the water clusters and alkane chains. Therefore, it is slightly difficult to see how impressive the method is or is not. It would also be very helpful if the authors would rework Figure 4. Since they reference Kuhne and co-workers paper, I looked at their paper to see the comparison but couldn’t figure out to compare to this paper. Kuhne plot goes from -0.3 to -1.8 kcal/mol. This plot goes from 132 to 399 meV. I’m fine with different units, but “zero” should be the same.

Minor thing:

Why was correlation energy defined as positive? It almost always is negative. But in figure 3 you obviously defined it is positive.

Summary: The idea really is good and I’m very enthusiastic about the story. However, at the moment I think it is hard to show exactly how transferable it is, and how accurate the proton transfer reaction free energy is. I think with some additional calculations it would show how transferable this ML methodology is and make it worthy of publication.

Reviewer #3 (Remarks to the Author):

The authors tried to build a mapping from electron density to correlation energy and demonstrated its applicability to some molecular systems. The idea is not new, the methods they used are already well-established in literature and their results are not that impressive. Therefore, I don't think it's worthy of publication in nat. comm. Instead, I'd suggest some specific journal like JCP or JPCA.

Here are some detailed comments.

To meet the standards of ncomm, the authors have to at least demonstrate that

i) their method is substantially better than the conventional approaches (no such comparison is made in the main text), i.e., mapping from geometry to correlation energy directly, e.g., a ML model built based on KRR or NN. I'm pessimistic about the superiority of such a test.

ii) The systems they dealt with are simple, if not totally trivial. To demonstrate the "remarkable possibilities" (as claimed in the abstract), I think their model has to reach similar accuracy for even larger systems, e.g., trained on $(\text{H}_2\text{O})_3$, application to $(\text{H}_2\text{O})_n$, with n reaching at least 8 to 12. The current illustration of scalability level is not impressive and the transferability is not that obvious without applicability to larger systems.

iii) the inherent drawback of the model is the need for charge density as input, which per se is not easy to obtain (in particular correlated charge density at high level of theory). To fully reap the conceptual advantage (as well as numerical superiority over dft or post-hf methods) of ML-DFA, the authors have to build a SCF model based on their current model. In principle, this is possible; but in practise I presume that a significant amount of training data would be needed, and the resulting model would be less favourable compared to the more conventional ML models. Once they can demonstrate such success, I'd be glad to consider their paper further.

iv) last but not least, there exists some fundamental drawbacks on the atom-centered expansion of charge density. Namely, it may fail to converge even if some big density-fitting basis is being used, due to the highly asymmetric nature of charge density for molecular systems. Actually, there are quite a lot of literature papers in the field of x-ray crystallography regarding the difficulties of reconstructing charge density through atom-centered basis functions. To be rigorous, the authors have to also do such convergence tests and discuss the limits of such approximations.

Responses to Reviewers of the Manuscript: Pure, non-local, machine-learned density functional theory for electron correlation

Johannes T. Margraf^{1, a)} and Karsten Reuter^{1, 2}

¹⁾*Chair for Theoretical Chemistry and Catalysis Research Center,
Technische Universität München, Lichtenbergstraße 4, D-85747 Garching,
Germany*

²⁾*Fritz-Haber-Institut der Max-Planck-Gesellschaft, Faradayweg 4-6,
D-14195 Berlin, Germany*

(Dated: 29 September 2020)

^{a)}Electronic mail: johannes.margraf@ch.tum.de

Reviewer 1 (Comments to the Author):

Reviewer 1: The method describes a solid piece of work with promising results. I am supportive of its publication. However, I do have concerns about whether the method is not a sufficient improvement upon existing methods and whether the numerical examples are sufficiently compelling to merit publication in Nat Commun.

Reply: We thank the reviewer for their encouraging comments. To underscore the potential of the proposed method, the numerical examples have been reworked. We now show results for transferability to systems which are twice as large as the original training set (e.g. $(\text{H}_2\text{O})_8$ and octane). In our view, these new results confirm the broad applicability of our approach (see below).

Reviewer 2 (Comments to the Author):

Reviewer 2: In the manuscript the authors discuss a machine learning method to accurately calculate the correlation energy from Hartree-Fock densities. The methodology is new, innovative, and accurate. This work is certainly worthy of publication; however, additional work maybe required to merit publication in Nature Communications.

Reply: We thank the reviewer for their positive assesment of our work.

Reviewer 2: 1. The authors are not performing DFT calculations their methodology is significantly more similar to post-HF methodologies. Particularly since they use the HF density. This methodology offers a way to get a correlation energy, like MP2 or CCSD(T). I think if the authors are going to motivate this work as a route to DFA, they need to show a methodology that would be self-consistent with the electron density. Right now, the methodology is a post-HF method, and it should be represented as such early in the paper.

I understand that density (or the density fitted basis) is the connection to the correlation energy. But right now, the difference between what is said in the abstract and what is actually done in the paper is large in my opinion.

Reply: We thank the reviewer for this thoughtful comment. Clearly, our method has many parallels with post-HF methods. However, we disagree with the assessment that we are not performing DFT calculations. We predict the correlation energy as a function of the electron density (and no other input). Our method is therefore a density functional. While DFT calculations are often performed self-consistently, this is by no means a requirement. Most calculations with higher-rung DFT functionals (i.e. RPA-based or double-hybrid methods) are performed non-selfconsistently. Similarly, vdW-functionals are very often applied as post-SCF corrections. Indeed, many functionals that are commonly used in SCF calculations were originally developed in a non-self-consistent framework (see e.g. the work of Axel Becke that led to the ubiquitous B3LYP functional among others).

We have expanded the discussion to include the post-HF perspective suggested by the reviewer, and to discuss the issue of self-consistency.

The new passage reads:

Finally, a note on self-consistency is in order. KS-DFT calculations are commonly carried out self-consistently. One could therefore argue that the presented approach is more closely related to post-HF methods like Coupled Cluster theory. Nonetheless, we feel that our models are more accurately described as DFAs, since they exclusively depend on the electron density. In particular, no information related to virtual orbitals is required. The models therefore share the favorable computational scaling of pure DFAs. We also note that higher-rung DFAs (i.e. double hybrids or RPA-based functionals) are still commonly used non-self-consistently.^{1,2} Indeed, even classic semilocal functionals like Becke’s B88 were developed in a non-self-consistent framework.³ In the same vein, a self-consistent extension of the proposed method is clearly possible and will be the subject of future work. This will, however, likely require significantly more training data.

Reviewer 2: 2. The first results they show are on water clusters and alkane chains. While they show transferability for correlation energy learned on propene and applied on

butane and suggest that this makes it applicable to large systems they don't show any larger systems. Since they are comparing to MP2, getting reference data for larger chain (even if they are not training on it) should be very feasible. I think a much stronger show of transferability would be to show significant longer alkane chains, octane for instance. Right now, I don't believe the authors should claim they have a transferable density functional if they can only show it can transfer from 3 water molecules to 4 and propane to butane. Even better show of transferability would be between the same atom in different environments (for instance ethane and ethene).

Reply: We agree that the results in the original manuscript did not fully explore the scope of transferability of the method. We therefore calculated new MP2 reference data for octane and H_2O_8 and validated the performance of models trained with up to four monomer units. We find good transferability of the models for this much more challenging test.

Fig. 3 and the corresponding discussion have therefore been replaced, as follows:

FIG. 1. Error histograms for predicted correlation energies E_C^{ML} of water octamer (left) and octane configurations (right) relative to MP2 reference energies E_C^{MP2} . The corresponding models were exclusively trained on systems with up to four monomers, exploiting the size-extensivity of the ML correlation functional.

In Fig. 1, this is highlighted for the water octamer and octane, with the corresponding ML-DFAs trained exclusively on the systems with up to four monomers (i.e. at most half the size of the predicted systems). Clearly, this is a challenging test for the transferability of the functionals. The observed MAEs are indeed somewhat larger than for models validated

on configurations of the same size as the training set (74 and 48 meV for $(\text{H}_2\text{O})_8$ and octane, respectively). A larger error should be expected for larger systems, however, precisely due to the extensive nature of the correlation energy.^{4,5} Furthermore, the prediction errors are somewhat systematic, in particular for the $(\text{H}_2\text{O})_8$. Consequently, the MAE for relative energies of water octamer configurations is further reduced to 42 meV (46 meV for octane).

Reviewer 2: 3. They only show results for 3 systems, water clusters, alkane chains, and a proton transfer. And they only truly have reference data for the water clusters and alkane chains. Therefore, it is slightly difficult to see how impressive the method is or is not. It would also be very helpful if the authors would rework Figure 4. Since they reference Kuhne and co-workers paper, I looked at their paper to see the comparison but couldnt figure out to compare to this paper. Kuhne plot goes from -0.3 to -1.8 kcal/mol. This plot goes from 132 to 399 meV. Im fine with different units, but zero should be the same.

Reply: While it is true that we only consider three different systems, these were judiciously chosen to cover a wide range of interactions, including covalent and hydrogen bonds, as well as van-der-Waals, polar and ionic interactions. As mentioned above, we have now included data on significantly larger systems, in order to provide true out-of-sample benchmarks for our method.

Regarding the scale of the free energies: In our case these are directly obtained from the negative log probability of a certain configuration appearing in the ensemble. Unfortunately, Kühne et al. do not provide detailed information on how their free energies are computed. Since they are all negative, these values have probably been shifted so that the largest free energy value is set to zero. Since our approach samples a broader range of configurations, doing the same would not lead to a comparable energy scale either, unfortunately. To facilitate future comparison, the full data underlying this figure is provided in the supporting information

Reviewer 2: Why was correlation energy defined as positive? It almost always is negative. But in figure 3 you obviously defined it is positive.

Reply: This figure has been replaced with the results on larger systems. We agree that the correlation energy is, of course negative. The previous plot showed the absolute value of the correlation energy.

Reviewer 2: Summary: The idea really is good and I'm very enthusiastic about the story. However, at the moment I think it is hard to show exactly how transferable it is, and how accurate the proton transfer reaction free energy is. I think with some additional calculations it would show how transferable this ML methodology is and make it worthy of publication.

Reply: We thank the reviewer for their positive feedback. In our view, the additional calculations on significantly larger systems provide a convincing proof of the transferability of our approach.

Reviewer 3 (Comments to the Author):

Reviewer 3: The authors tried to build a mapping from electron density to correlation energy and demonstrated its applicability to some molecular systems. The idea is not new, the methods they used are already well-established in literature and their results are not that impressive. Therefore, I don't think it's worthy of publication in nat. comm. Instead, I'd suggest some specific journal like JCP or JPCA.

Reply: We thank the reviewer for their feedback. However, we would appreciate some more specific information as to where the ideas and methods presented in the paper have previously been reported. The discussion section of the manuscript aims to provide a comprehensive perspective on how our approach compares to the state-of-the-art. In our view

(and also reflected by the statements of the other reviewers), this underscores the novelty of the presented approach.

Reviewer 3: To meet the standards of ncomm, the authors have to at least demonstrate that i) their method is substantially better than the conventional approaches (no such comparison is made in the main text), i.e., mapping from geometry to correlation energy directly, e.g., a ML model built based on KRR or NN. I’m pessimistic about the superiority of such a test.

Reply: As the reviewer notes, geometry based ML models are now commonplace in chemistry and have achieved impressive accuracy. Compared to our proposed method, these models have the advantage that they do not require a previous electronic structure calculation to obtain the electron density. However, a geometry based prediction of the correlation energy is not very useful, in our opinion. Since only the sum of the Hartree-Fock and correlation energies (the total energy) is chemically meaningful, a previous electronic structure calculation is required regardless of whether a geometric or density based representation is used.

The question then is: Does our approach outperform a geometry-based prediction of the total energy? While a comprehensive analysis of different geometric ML methods is beyond the scope of this work, we now provide an example of this with the state-of-the-art Many-Body-Tensor Representation (MBTR) of Huo and Rupp (in the revised SI). While the MBTR can be used to fit an accurate model of the total energy, this requires much more data than the ML-DFA approach (to achieve comparable accuracy).

The new passage in the SI reads:

The overwhelming majority of ML applications in chemistry use a representation of the molecular geometry as input.⁶ Compared to the proposed density functional approach, this has the advantage that no electronic structure (e.g. Hartree-Fock) calculations have to be performed at prediction time to obtain the electron density. On the other hand, the ML-DFA method takes advantage of the fact that the Hartree-Fock energy is automatically computed

FIG. 2. Comparison of a geometry based MBTR model (orange) and a ML-DFA (blue) for the water tetramer.

in addition to the density, so that only the correlation energy has to be learned from data. Overall, one would therefore expect a trade-off: to reach the same accuracy, a ML-DFA will need fewer training examples than a geometry-based ML model, but it will be more expensive to evaluate at prediction time. If the reference data is calculated with a high-level reference method (e.g. coupled cluster theory) and the target systems are small enough that a mean-field electronic structure calculation (e.g. at the HF level) is affordable, this trade-off favors the ML-DFA. In contrast, a model trained on DFT data and applied to very large systems would clearly be more efficient using a geometric representation.

There are a wide range of geometric representations and ML models, a comprehensive assessment of which is beyond the scope of the current work. However, to provide quantitative support for the above discussion, we fitted a KRR model for the water tetramer using the Many-Body-Tensor Representation (MBTR) of Huo and Rupp as a representative example of state-of-the-art geometry-based ML.⁷ As can be seen in Fig. 2, the MAE for the geometry based model is on average an order of magnitude larger than for the ML-DFA model. From a different perspective, to reach a similar accuracy as the ML-DFA, one requires an order of magnitude more training examples.

Reviewer 3: ii) The systems they dealt with are simple, if not totally trivial. To

demonstrate the "remarkable possibilities" (as claimed in the abstract), I think their model has to reach similar accuracy for even larger systems, e.g., trained on (H₂O)₃, application to (H₂O)_n, with n reaching at least 8 to 12. The current illustration of scalability level is not impressive and the transferability is not that obvious without applicability to larger systems.

Reply: As discussed in the response to reviewer 2, new data has been included in the revised manuscript, which shows the transferability of the models to octane and the water octamer.

Reviewer 3: iii) the inherent drawback of the model is the need for charge density as input, which per se is not easy to obtain (in particular correlated charge density at high level of theory). To fully reap the conceptual advantage (as well as numerical superiority over dft or post-hf methods) of ML-DFA, the authors have to build a SCF model based on their current model. In principle, this is possible; but in practise I presume that a significant amount of training data would be needed, and the resulting model would be less favourable compared to the more conventional ML models. Once they can demonstrate such success, I'd be glad to consider their paper further.

Reply: We agree that a self-consistent model would fully realize the potential of the ML-DFA approach. As the reviewer mentions, this would require significantly more data and theoretical development, which are beyond the scope of the current work. As discussed above, we do not believe that self-consistency is inherently a requirement for density functional theory. Furthermore, our numerical results show that the non-self-consistent ML-DFA presented herein is highly accurate and data efficient, which makes it interesting in its own right. The revised discussion of self-consistency (see above) reads:

Finally, a note on self-consistency is in order. KS-DFT calculations are commonly carried out self-consistently. One could therefore argue that the presented approach is more closely related to post-HF methods like Coupled Cluster theory. Nonetheless, we feel that

our models are more accurately described as DFAs, since they exclusively depend on the electron density. In particular, no information related to virtual orbitals is required. The models therefore share the favorable computational scaling of pure DFAs. We also note that higher-rung DFAs (i.e. double hybrids or RPA-based functionals) are still commonly used non-self-consistently.^{1,2} Indeed, even classic semilocal functionals like Becke’s B88 were developed in a non-self-consistent framework.³ In the same vein, a self-consistent extension of the proposed method is clearly possible and will be the subject of future work. This will, however, likely require significantly more training data.

Reviewer 3: iv) last but not least, there exists some fundamental drawbacks on the atom-centered expansion of charge density. Namely, it may fail to converge even if some big density-fitting basis is being used, due to the highly asymmetric nature of charge density for molecular systems. Actually, there are quite a lot of literature papers in the field of x-ray crystallography regarding the difficulties of reconstructing charge density through atom-centered basis functions. To be rigorous, the authors have to also do such convergence tests and discuss the limits of such approximations.

Reply: While we appreciate this comment, we do not see how it is relevant to the current work. Density fitting basis sets have been used in virtually all molecular electronic structure codes for many years. The accuracy of the density fitting approximation has been firmly established in this context. In such calculations, the density is already expanded in an atom-centered basis-set, which is a completely different situation than in X-ray crystallography, where one works with noisy experimental data.

REFERENCES

- ¹S. Grimme, *J. Chem. Phys.* **124**, 034108 (2006).
- ²I. Y. Zhang, P. Rinke, J. P. Perdew, and M. Scheffler, *Phys. Rev. Lett.* **117**, 133002 (2016).
- ³A. D. Becke, *Phys. Rev. A* **38**, 3098 (1988).
- ⁴J. P. Perdew, J. Sun, A. J. Garza, and G. E. Scuseria, *Zeitschrift für Phys. Chemie* **230**, 737 (2016).
- ⁵J. T. Margraf, D. S. Ranasinghe, and R. J. Bartlett, *Phys. Chem. Chem. Phys.* **19**, 9798 (2017).
- ⁶O. A. von Lilienfeld, *Angew. Chemie Int. Ed.* **57**, 4164 (2018).
- ⁷H. Huo and M. Rupp, (2017), arXiv:1704.06439.

REVIEWERS' COMMENTS

Reviewer #2 (Remarks to the Author):

In the revised manuscript the authors have address my concerns. While I still feel this is more of a post-HF method, it is semantics. I believe the updated results with larger water clusters and long alkene chains make the demonstration of transferability significantly more clear. Additionally, I believe the point made in the SI – that the training data required for similar accuracy between this approach and a geometry only approach is significantly different is a great addition to the paper. I believe the manuscript is now suitable for publication.

Reviewer #3 (Remarks to the Author):

Reviewer 3: The authors tried to build a mapping from electron density to correlation energy and demonstrated its applicability to some molecular systems. The idea is not new, the methods they used are already well-established in literature and their results are not that impressive. Therefore, I don't think it's worthy of publication in nat. comm. Instead, I'd suggest some specific journal like JCP or JPCA.

Reply: We thank the reviewer for their feedback. However, we would appreciate some more specific information as to where the ideas and methods presented in the paper have previously been reported. The discussion section of the manuscript aims to provide a comprehensive perspective on how our approach compares to the state-of-the-art. In our view (and also reflected by the statements of the other reviewers), this underscores the novelty of the presented approach

Response:

Title and abstract of this ms read as if the authors are the first to use ML to predict correlation based on density.

This is not true and as such the authors are, consciously or not, attempting to create that (admittedly appealing) impression.

It is true that the authors refer to the relevant literature (in particular work by Miller and co-workers, and Burke

and co-workers) and explicitly discuss the differences to their approach.

Components of the authors' contribution are most certainly new which is why I had suggested publication in

JCP or JPCA which are very fine journals for such technical progress.

Regarding Miller's work, they explicitly state the following:

"The main difference is that in our case no previous energy decomposition or orbital localization is required.

This enables us to choose arbitrary reference methods, for which pair correlation energies may not be available

or implemented (i.e. the random phase approximation, RPA, or quantum Monte Carlo methods)."

In other words, the authors admit themselves that they are not the first building density based ML models of correlation, and that

rather they can circumvent the localization step. While the need for localization might be cumbersome (especially if not 'implemented'

as the authors say), this is obviously not a major conceptual scientific breakthrough but rather an incremental advantage

(there are many powerful localization methods and the fact that something no longer needs to be implemented

should not warrant a paper in Nature Communications).

Thereafter the authors write:

"Furthermore, the requirement of orbital localization makes the application of MOB-ML to metallic or small band-gap systems fundamentally difficult."

Indeed, this could constitute a more valid reason that the author's approach actually bears some major significance in terms of increasing the applicability to metallic systems. Regrettably though, the authors do not demonstrate their approach for such systems! As such, this 'bashing' on other approaches to make their method shine remains disappointingly unsubstantiated.

Regarding Burke's work the authors state:

"The main difference to our work is their choice of density representation. Specifically, they work in a plane-wave basis instead of the atom-centered Gaussians used herein. This makes learning the density much easier, since all basis functions are orthogonal. As a consequence of this choice, their models are not size-extensive, however."

Again, the authors admit that they are not the first doing this but rather that they are simply using a atom-centered Gaussians instead of plane-waves (with the obvious advantage (they omit the disadvantages) that they are size consistent by construction.

Now, revisiting a published idea with a different set of basis-functions is technically interesting but scientifically rather incremental. JCP or JPCA frequently publish such work which is why I suggested them.

Reviewer 3: To meet the standards of ncomm, the authors have to at least demonstrate that i) their method is substantially better than the conventional approaches (no such comparison is made in the main text), i.e., mapping from geometry to correlation energy directly, e.g., a ML model built based on KRR or NN. I'm pessimistic about the superiority of such a test.

Reply: As the reviewer notes, geometry based ML models are now commonplace in chemistry and have achieved impressive accuracy. Compared to our proposed method, these models have the advantage that they do not require a previous electronic structure calculation to obtain the electron density. However, a geometry based prediction of the correlation energy is not very useful, in our opinion. Since only the sum of the Hartree-Fock and correlation energies (the total energy) is chemically meaningful, a previous electronic structure calculation is required regardless of whether a geometric or density based representation is used.

The question then is: Does our approach outperform a geometry-based prediction of the total energy? While a comprehensive analysis of different geometric ML methods is beyond the scope of this work, we now provide an example of this with the state-of-the-art Many-Body-Tensor Representation (MBTR) of Huo and Rupp (in the revised SI). While the MBTR can be used to fit an accurate model of the total energy, this requires much more data than the ML-DFA approach (to achieve comparable accuracy).

Response:

The comparison to MBTR is meaningful and certainly improves this contribution.

Inspection of the new figure (Fig. 2 in the report) makes me wonder about the validity of the authors' conclusions.

(a) As everyone can see, the learning curve for the blue dots indicates hardly any learning, hovering above 0.01 eV from 50 to 1000 data points.

The orange dots, however, nicely improve with training set size reaching nearly 0.2 eV for N=1000. As such, the trade-off clearly depends on training set size and desirable prediction error.

(b) The comparison is unfair since training data set corresponds to a different cost for the two methods.

As such, the authors should report error as a function of acquisition cost for training data.

In common quantum chemistry methods for molecular systems, the cost for the HF calculation dominates,

and one would therefore expect the MBTR learning curve to be substantially below the ML-DFA curve.

(c) MBTR is not the most accurate structure based approach in the field. Comparison to a state-of-the-art

approach would be more appropriate.

(d) Even if the authors were correct in their assumption (which is being defied in above points) that training

ML-DFA is more cost-efficient than MBTR - this hardly matters. It is the entire point of using machine learning

that the computational expense is done beforehand in order to accelerate the prediction speed.

And, as the authors

acknowledge themselves, they will always have to include a HF calculation in the cost of their predictions.

By contrast, MBTR predictions, once properly trained, are practically free.

As such, I do not find it obvious at all that the authors' claim for superiority is warranted.

Their approach is clearly interesting on a technical level, but it is unlikely to dramatically advance the field.

Responses to Reviewers of the Manuscript: Pure, non-local, machine-learned density functional theory for electron correlation

Johannes T. Margraf^{1, a)} and Karsten Reuter^{1, 2}

¹⁾*Chair for Theoretical Chemistry and Catalysis Research Center,
Technische Universität München, Lichtenbergstraße 4, D-85747 Garching,
Germany*

²⁾*Fritz-Haber-Institut der Max-Planck-Gesellschaft, Faradayweg 4-6,
D-14195 Berlin, Germany*

(Dated: 23 November 2020)

^{a)}Electronic mail: johannes.margraf@ch.tum.de

Reviewer 2 (Comments to the Author):

Reviewer 2: In the revised manuscript the authors have address my concerns. While I still feel this is more of a post-HF method, it is semantics. I believe the updated results with larger water clusters and long alkene chains make the demonstration of transferability significantly more clear. Additionally, I believe the point made in the SI that the training data required for similar accuracy between this approach and a geometry only approach is significantly different is a great addition to the paper. I believe the manuscript is now suitable for publication.

Reply: We thank the reviewer for their encouraging feedback.

Reviewer 3 (Comments to the Author):

Reviewer 3: Title and abstract of this ms read as if the authors are the first to use ML to predict correlation based on density. This is not true and as such the authors are, consciously or not, attempting to create that (admittedly appealing) impression. It is true that the authors refer to the relevant literature (in particular work by Miller and co-workers, and Burke and co-workers) and explicitly discuss the differences to their approach. Components of the authors' contribution are most certainly new which is why I had suggested publication in JCP or JPCA which are very fine journals for such technical progress.

Reply: We thank the reviewer for their feedback. Obviously it is not our intention to downplay the quality or relevance of the work of Miller and Burke, which is why we extensively discuss it in the paper. The title of our manuscript ("Pure, non-local, machine-learned density functional theory for electron correlation") is simply a description of its content and does not claim to be the first work to use ML in the context of DFT. Admittedly, the use of the generic acronym ML-DFA for our approach may raise that impression, however. To avoid this, we have revised the manuscript and refer to our models as Kernel Density Functional Approximations (KDFA), whereas we still use ML-DFA for the general concept of ML based DFT approximations. We accordingly revised the abstract, which now reads:

Here, we present a type of machine-learning (ML) based DFA (termed Kernel Density Functional Approximation, KDF) that is pure, non-local and transferable, and can be efficiently trained with fully quantitative reference methods.

Reviewer 3: Regarding Miller’s work, they explicitly state the following: ”The main difference is that in our case no previous energy decomposition or orbital localization is required. This enables us to choose arbitrary reference methods, for which pair correlation energies may not be available or implemented (i.e. the random phase approximation, RPA, or quantum Monte Carlo methods).” In other words, the authors admit themselves that they are not the first building density based ML models of correlation, and that rather they can circumvent the localization step. While the need for localization might be cumbersome (especially if not ’implemented’ as the authors say), this is obviously not a major conceptual scientific breakthrough but rather an incremental advantage (there are many powerful localization methods and the fact that something no longer needs to be implemented should not warrant a paper in Nature Communications).

Thereafter the authors write: ”Furthermore, the requirement of orbital localization makes the application of MOB-ML to metallic or small band-gap systems fundamentally difficult.” Indeed, this could constitute a more valid reason that the author’s approach actually bears some major significance in terms of increasing the applicability to metallic systems. Regrettably though, the authors do not demonstrate their approach for such systems! As such, this ‘bashing’ on other approaches to make their method shine remains disappointingly unsubstantiated.

Reply: The approach of Miller and co-workers is not based on the electron density, but uses information from occupied and virtual molecular orbitals. Regarding the need for localization: We agree that there are many powerful orbital localization methods, which are available in many software packages. However, not all post-HF methods can be formulated in a local MO basis (hence the examples of QMC and most variants of RPA, for which this

is not the case). This is not just a technical issue, but the scientific formalism that allows such energy decomposition simply does not exist.

We appreciate that the referee found the argument with respect to small-band gap systems persuasive. We agree that we provide no examples of this in the paper, which is why it is part of the discussion and not the results section. It is certainly not our intention to bash anyone, in this context. This is simply a factual statement that is relevant to the discussion.

Reviewer 3: Regarding Burke’s work the authors state: ”The main difference to our work is their choice of density representation. Specifically, they work in a plane-wave basis instead of the atom-centered Gaussians used herein. This makes learning the density much easier, since all basis functions are orthogonal. As a consequence of this choice, their models are not size-extensive, however.” Again, the authors admit that they are not the first doing this but rather that they are simply using a atom-centered Gaussians instead of plane-waves (with the obvious advantage (they omit the disadvantages) that they are size consistent by construction. Now, revisiting a published idea with a different set of basis-functions is technically interesting but scientifically rather incremental. JCP or JPCA frequently publish such work which is why I suggested them.

Reply: In the comparison of plane-waves and Gaussians we state one advantage (orthogonality) and one disadvantage (lack of size-extensivity) of the former. In our view size-consistency and extensivity are not technical details, however, as they fundamentally change the types of applications a method can handle. Indeed, size-consistency is one of the defining features of a useful model chemistry according to John Pople.¹ We therefore feel that the importance of this change goes beyond a mere technical improvement.

Reviewer 3: The comparison to MBTR is meaningful and certainly improves this contribution. Inspection of the new figure (Fig. 2 in the report) makes me wonder about the

validity of the authors' conclusions. (a) As everyone can see, the learning curve for the blue dots indicates hardly any learning, hovering above 0.01 eV from 50 to 1000 data points. The orange dots, however, nicely improve with training set size reaching nearly 0.2 eV for $N=1000$. As such, the trade-off clearly depends on training set size and desirable prediction error. (b) The comparison is unfair since training data set corresponds to a different cost for the two methods. As such, the authors should report error as a function of acquisition cost for training data. In common quantum chemistry methods for molecular systems, the cost for the HF calculation dominates, and one would therefore expect the MBTR learning curve to be substantially below the ML-DFA curve. (c) MBTR is not the most accurate structure based approach in the field. Comparison to a state-of-the art approach would be more appropriate. (d) Even if the authors were correct in their assumption (which is being defied in above points) that training ML-DFA is more cost-efficient than MBTR - this hardly matters. It is the entire point of using machine learning that the computational expense is done beforehand in order to accelerate the prediction speed. And, as the authors acknowledge themselves, they will always have to include a HF calculation in the cost of their predictions. By contrast, MBTR predictions, once properly trained, are practically free.

As such, I do not find it obvious at all that the authors' claim for superiority is warranted. Their approach is clearly interesting on a technical level, but it is unlikely to dramatically advance the field.

Reply: We appreciate the detailed feedback provided by the referee and provide a point-by-point response to their concerns in the following:

(a) The learning rate for the ML-DFA is indeed lower than for MBTR. We also agree that there is a trade-off with respect to the training set size, as we explicitly discuss in the SI. These observations are not in conflict with our conclusions, however. Most importantly, the ML-DFA error is still orders of magnitude lower than MBTR for small training sets.

(b) We disagree with this statement. The acquisition cost for the training data is identical for both methods (it is the same data). Furthermore it is incorrect that the cost of the HF calculation dominates in common quantum chemistry methods for molecular systems. This is only true for MP2 with small systems and basis sets. For CCSD(T) the CC calculation will almost always dominate the HF one significantly. This is especially true in the asymptotic

limit of large systems and basis-sets.

(c) In our view a benchmark of different geometrical ML methods is beyond the scope of this paper. All data for reproducing the plot with a different ML representation is provided.

(d) We agree that the point of ML is to reduce the cost of prediction at the expense of training data. In our view, this does not mean that the cost of generating training data is unimportant. We show that for a given target accuracy, there is a trade-off between how much the prediction time is reduced and how costly the generation of the training set is.

REFERENCES

¹Pople, J. A. Nobel Lecture: Quantum chemical models. *Rev. Mod. Phys.* **71**, 1267–1274 (1999).